# Vaccination and Trust in the National Health System among HIV+ Patients: An Italian Cross-Sectional Survey

**DOI:** 10.3390/vaccines11081315

**Published:** 2023-08-02

**Authors:** Fabrizio Bert, Antonino Russotto, Alex Pivi, Benedetta Mollero, Gianluca Voglino, Giancarlo Orofino, Roberta Siliquini

**Affiliations:** 1Department of Public Health and Pediatrics Sciences, University of Turin, 10126 Turin, Italy; 2Hygiene and Infection Control Unit, ASL TO3, 10098 Turin, Italy; 3School of Medicine, University of Turin, 10126 Turin, Italy; 4Ministry of Health, 00144 Rome, Italy; 5Amedeo di Savoia Hospital, 10149 Turin, Italy; 6Hospital Molinette, City of Health and Science of Turin, 10126 Turin, Italy

**Keywords:** vaccines, HIV, infectious diseases

## Abstract

Background: This study aimed to evaluate the knowledge, attitudes, and practises (KAP) of individuals living with HIV (PLWH) regarding vaccines and their trust in the Italian national health system (NHS). Methods: A cross-sectional survey was conducted at Amedeo di Savoia Hospital in Turin, Italy, involving 160 HIV-positive patients. Descriptive statistics were utilised to analyse variables such as vaccination status and intention, perceived risk of infection, and disease severity. The infections were categorised into sexually transmitted diseases and other vaccine-preventable diseases. Results: Except for the perceived severity of infection, there were no significant differences in the percentages between the two infection groups for the variables examined. Concerning patients’ perception of the Italian NHS, a high percentage of the sample believed in the information provided by healthcare workers (HCWs) (95.6%) and considered HCWs up-to-date on vaccines (93.1%). However, a considerable proportion expressed concerns about insufficient information on vaccine risks from HCWs (33.3%), perceived judgement by HCWs for vaccine refusal (40.3%), and suspected financial interests of HCWs in vaccination (19.5%). Conclusions: Some HIV+ patients may hesitate to be vaccinated or hold misconceptions about the severity of certain infectious diseases. Additionally, there are concerns about trust in the Italian NHS and communication by HCWs.

## 1. Introduction

Vaccines are one of the most relevant public health tools to determine the decline in morbidity and mortality related to various infectious diseases; as a prime example, the eradication of smallpox worldwide [1] or the elimination of poliomyelitis in the Americas [2]. In the last few years, however, people have started questioning incrementally the need for and safety of vaccines, challenging the medical community’s ability to maintain high vaccination rates in certain communities [3]. Subsequently, the concept of Vaccine Hesitancy, defined as “a delay in acceptance or refusal of vaccination despite the availability of vaccination services”, has begun to appear in the scientific landscape [3,4]. In this sense, a systematic review conducted from 2004 to 2014 highlighted how, at the European level, many people often believe that the risks of vaccination outweigh their benefits [5]. This uncertainty has caused a decrease in vaccinations and an increase in cases of some vaccine-preventable infectious diseases, such as seasonal Influenza and other infectious diseases, in the European Union [6,7]. As happened for measles in 2019, when the largest number of cases in a single year since 1994 were reported in the United States, despite measles being declared eliminated in 2000 [8]. Vaccine hesitancy in 2019 was identified as one of the 10 threats to global health by the World Health Organisation [9]. As reported by the WHO definition of hesitancy, it “is complex and context-specific, varying across time, place, and vaccines. It is influenced by factors such as complacency, convenience, and confidence” [10]. Vaccine hesitancy varies depending on certain variables, such as socio-demographic variables (e.g., number of years of schooling or age), political motivations (e.g., vaccine policies/programmes), characteristics of the healthcare system (e.g., paid/free vaccines), individuals’ perceived risk of vaccine-preventable diseases (e.g., a limited knowledge of vaccine-preventable diseases), and knowledge of vaccine-preventable diseases [11,12,13,14]. These results are summarised in a meta-analysis, which highlighted how risk perception is a predictor of vaccination behaviour [15]. Vaccine hesitancy can then vary according to the vaccine involved (one can be hesitant regarding the flu vaccine but accept with confidence all other vaccines), with newer vaccines usually engendering more hesitancy [16], or it can vary according to specific target populations such as patients with chronic diseases of various types [17], as discussed in our recent study [18]. The population affected by chronic pathologies and immune system dysfunctions certainly includes people living with HIV (PLWH). This pathological condition determines an increased risk of contracting infectious diseases and increased clinical severity compared to non-HIV-positive subjects [19,20]. Although several studies have not found a significant association, it is possible that the level of knowledge about the disease in PLWH influences the perceived risk and that, therefore, a low level of knowledge corresponds to a low perception of risk [21]. In this sense, a study has shown that in PLWH who are vulnerable from a socio-economic point of view, a reduction in the vaccination adherence rate is observed [22]. In general, several studies have reported inadequate rates of adherence to vaccination campaigns by PLWH [22,23,24,25].

Exploring the adherence to vaccination campaigns among individuals is of fundamental importance due to the potential public health repercussions that any deficiencies in protection among these individuals may have. Although it concerns a selected segment of the population, analysing their vaccine propensity in general and across different nations contributes to identifying the real-world framework of the vaccine hesitancy phenomenon. Such analysis serves as the starting point for building effective preventive strategies to combat major infectious diseases within a group of individuals deserving special attention.

In the literature, there are no in-depth studies in Italy that evaluate vaccination coverage and measure the phenomenon of vaccination hesitancy in a specific population, such as that of HIV-positive patients, except for one study [18], which nevertheless is focused only on COVID-19 vaccination. Data regarding vaccine hesitancy towards the COVID-19 vaccine, however, may be different given the pandemic situation from data regarding other vaccines. Given the clinical conditions of PLWHs, it becomes even more crucial to carry out this study. The main purpose of this study was to evaluate the knowledge, attitudes, and practises (KAP) of HIV-positive subjects in terms of vaccination, thus excluding vaccination against COVID-19, in order to reflect on possible vaccine hesitancy and on potential communication strategies that could improve vaccination adherence. This study is part of the framework of a large national project funded by the Italian National Recovery and Resilience Plan (PNRR), which dedicates an entire work package to the analysis of the vaccine hesitancy phenomenon in the general population and specific subgroups.

## 2. Materials and Methods

### 2.1. Study Design

From November 2021 to April 2022, a cross-sectional survey was conducted at the infectious disease clinic of Amedeo di Savoia Hospital in Turin. The Amedeo di Savoia Hospital is the regional reference centre (in the Piemonte region) for the diagnosis and treatment of HIV and related pathologies. For this reason, there is a large caseload of HIV-positive individuals who are being cared for at a dedicated clinic. The study involved PLWH and consisted of 160 questionnaires (Appendix A). To be eligible, participants had to meet the following criteria: age greater than 18 years old, laboratory-confirmed HIV diagnosis, not being a first-time visitor to the clinic, and having the ability and willingness to understand the study. Prior to participation, all participants provided informed consent after receiving detailed information about the study’s purpose and objectives. The Ethics Committee of Azienda Ospedaliero-Universitaria and the City of Health and Science of Turin approved all procedures conducted in this study (protocol code n. 0091591, 23 September 2019).

### 2.2. Data Collection

Data collection took place at the infectious disease clinic mentioned above in Turin. The questionnaire consisted of two sections with a total of 40 items. The first section focused on socio-demographic variables, while the second section covered patients’ vaccination records, their intention to receive vaccinations, and their knowledge regarding major infectious diseases preventable through vaccination. The list of infectious diseases examined included: Hepatitis A, Hepatitis B, Tetanus (booster), Diphteria, Pertussis, HPV, Pneumococcus, Influenza, Varicella, Measles, Mumps, Rubella, Poliomyelitis, Meningococcus, Type B Heamophilus Influenzae, and Herpes Zoster. We have chosen these microorganisms because they are all part of the national vaccination prevention programme in Italy, and, therefore, their relative vaccines are all potentially recommended for the protection of the health of the immunocompromised population.

As described in our previously published article [18], each patient completed both parts of the questionnaire independently, with the option to seek clarification from a researcher present in the same room. All information collected during the study was handled according to the procedures outlined in our previously published article [18]. Informed consent was obtained separately from the questionnaire, and the consent forms are stored independently from the questionnaires and are accessible only to the researchers conducting the study. All questionnaires were anonymously completed from the outset. Researchers cannot in any way link individual questionnaires to the participant’s name.

### 2.3. Statistical Analysis

Following data collection, four researchers anonymously entered the data into an informatics database located in the Department of Public Health and Paediatrics at the University of Turin. Descriptive analyses were performed for all variables. Continuous variables were presented as mean ± SD, while categorical variables were expressed as percentages. Missing values were excluded. Variables pertaining to vaccination status, vaccination intention, perceived risk of infection, and disease severity were analysed by computing the means. Infections were categorised into two groups. The first group encompassed sexually transmitted diseases (STDs): hepatitis A (HVA), hepatitis B (HVB), and Human Papilloma Virus (HPV), while the second group included other types of infections such as Tetanus, Diphtheria, Pertussis, Pneumococcus, Influenza, Varicella, Measles, Mumps, Rubella, Poliomyelitis, Meningococcus, Type B Haemophilus Influenzae, and Herpes Zoster. This division aimed to explore potential differences in attitudes towards the STD group compared to the other group, considering that 83.5% of HIV diagnoses in Italy are associated with unprotected sex [26].

Finally, a confidence score towards the NHS (National Health Service) was calculated by assigning one point to responses indicating trust. The total score is the sum of individual scores and can range from 0 to 7, where 7 represents the highest level of trust. To evaluate differences in the confidence score across categories defined by vaccination status, the Kruskal-Wallis or Mann-Whitney test was performed where appropriate.

## 3. Results

The socio-demographic and characteristic analysis of the patients involved in this study was provided in Table 1 of our previous published study [18]. In this study, therefore, we only propose the main relevant data. The median age of the sample was 49.97 (SD ± 11.82) years; most (69.38%) of the patients had no chronic diseases (other than HIV); 62.68% had a lymphocyte count ≥ 501; and 79.33% had an undetectable viral load. On average, participants had been living with HIV for 12.03 years (SD ± 9.14).

The sample’s vaccination status was described in Table 1. Vaccination’s rate percentages in the sexually transmitted infection group (51.16%) did not differ particularly from the second group (48.58%). In particular, low coverage has emerged concerning HPV vaccination, with only 44.16% of the sample vaccinated. Adding up the percentages of natural immunity and vaccination, low coverage has emerged also forInfluenza (34.42%), Poliomyelitis (38.96%), rubella (44.81%), and mumps (50%). Interestingly, Type B Haemophilus Influenzae vaccination status was unknown by almost half of the sample. The analysis of the overall situation highlights a need to strengthen the current vaccination strategy, considering the reported percentages of non-vaccination (ranging from 11.61% to 50.00% depending on the different vaccines) and the limited knowledge of one’s vaccination status (between 16.88% and 48.37% of responses depending on the vaccine considered).

The sample’s vaccination intention is covered in Table 2. The willingness to be vaccinated (if not already immune) was 14.76% for the sexually transmitted infection group and 20.61% for the second group. Interestingly, almost one in four patients is in doubt about whether they want to be vaccinated against HPV, and 16.99% refuse the vaccination. Similar data, provided in Table 2, was obtained for mumps and rubella. The highest vaccination refusal rate has emerged for the Influenza virus (32.09%) and Type B Haemophilus Influenzae (23.18%). Instead, the greatest willingness to be vaccinated was obtained for shingles (38.61%). In many cases, for instance, in 33.11% of the cases for type B Haemophilus Influenzae or 26.14% for rubella, the question about vaccination intention receives the response “I do not know”. This data provides significant food for thought regarding the opportunities to raise awareness among these individuals about the importance of vaccination to reduce their chances of falling into the hesitant category.

Percentages about the perception of infection’s risk and the perception of infection severity were discussed in Table 3.

The highest perceived risk of infection rate was obtained for Influenza (49.35%) and Herpes Zoster (38.31%). The data regarding the flu is not surprising, considering that it is the condition with which the participants have most likely and frequently dealt over the years.

Instead, the highest perceived risk of infection severity rate was for HBV (87.58%), Meningococcus (86.18%), and, interestingly, Type B Haemophilus Influenzae (83.66%). It is interesting to note how two of the most feared pathogens in terms of the severity of the disease are among those responsible for meningitis, considered probably one of the most dangerous complications of infection not only in this subgroup of individuals but also in the general population.

The perception of the risk of infection transmission for some microorganisms is evidently not correct in this sample. For instance, Measles (high risk 14.38%) and Varicella (high risk 13.07%) are among the most contagious viruses and are more likely to cause epidemics in cases of low vaccination rates. Similarly, some diseases are erroneously considered not severe (i.e., Varicella, low severity (52.60%), and Measles, low severity (49.35%), despite being capable of leading to very serious or lethal complications.

Trust in the Italian NHS was addressed in Table 4. Although positive opinions about the information given by healthcare workers and their updated knowledge were shared by almost the entire sample, alarming results have emerged on other issues. The perception of being blamed in cases of refusal to vaccinate was observed in 40% of the sample; the opinion that HCWs have financial interests in vaccinating was shared by 19.50% of the respondents; and in addition, 33.33% of the sample believed that HCWs do not give enough information about the risks related to vaccination.

Luckily, the majority of respondents (95.63%) declared to be confident in the information provided by NHS workers and think that HCWs are prepared and up-to-date on vaccines (93.08%). In one out of five cases, a logistical difficulty in approaching vaccination is highlighted (“I disagree that the organisation of the vaccination offer is flexible in terms of timing and methods”), identifying convenience, understood as accessibility to vaccination, as a determinant of vaccine hesitancy.

The confidence score had a median of 6 (IQR = 5–7). For most vaccines, there was no significant relationship between vaccination status and the score, except for the following: Measles (medians for unvaccinated vs. vaccinated were 4.5 vs. 6; *p* = 0.027), Meningococcus (medians for unvaccinated vs. vaccinated were 5 vs. 6; *p* = 0.002), and COVID-19 (medians for unvaccinated vs. vaccinated were 5 vs. 6; *p* = 0.048). The correlation between low trust in the national healthcare system and measles vaccination could reflect the persistent fake news that has been promoted for years by anti-vaccine groups regarding the alleged and widely debunked association between the measles vaccine and autism. This association has been hypothesised but extensively refuted by numerous studies in the literature.

## 4. Discussion

To date, in the absence of further updates, the Italian guidelines suggest administering the following vaccines in the HIV-positive population: anti-influenza, anti-hepatitis B, anti-meningococcal, anti-pneumococcal, anti-haemophilus, and anti-zoster vaccines. Measles-mumps-rubella and varicella vaccines are recommended only in the presence of a CD4 lymphocyte count ≥ 200/mL [27]. In addition, the National Intervention Plan against HIV and AIDS (PNAIDS) suggested the achievement of a target of 70% implementation of suggested vaccinations [28]. It is interesting to note that none of the vaccines mentioned above reached this target in our study. 

In this paper, the available vaccines were divided into two groups: the first was formed by sexually transmitted infections (HVA, HVB, and HPV), and the second by the remaining (Tetanus, Diphtheria, Pertussis, Pneumococcus, Influenza, Varicella, Measles, Mumps, Rubella, Poliomyelitis, Meningococcus, Type B Haemophilus Influenzae, and Herpes Zoster Virus). Regarding vaccination against hepatitis A, the vaccination coverage in our study (48.4%) was in line with a study conducted in France (47.4%) [29]. Including natural immunity, the percentage of immune subjects (51.6%) was in line with another study conducted in the United States at baseline (58%) [30]. As regards vaccination against hepatitis B, the vaccination coverage in our study (53.2%) was higher than in a Belgium study (24.4%) [31] and in line with what was reported by other studies conducted in Brazil and France (57.3% and 61.9%) [29,30,31,32]. It is interesting to note instead how the vaccination coverage against HPV (44.2%) was significantly lower than in a Mexican study (90.1%) [33]. 

It is also interesting to highlight how, although vaccination against herpes zoster with Recombinant Zoster Vaccine (RZV) was authorised only on 30 September 2021, in Piedmont [34], and that the window period of our study was from November 2021 to April 2022, a good adherence/willingness rate for vaccination was found. In fact, although only 13.2% of our sample declared having been vaccinated against herpes zoster, it is also true that 39.6% declared that they intended to undergo the vaccination or had already planned it for the following weeks. If this willingness to get vaccinated is confirmed, overall vaccination coverage against herpes zoster could be higher than in other studies [24,25]; this could highlight a good specific communication campaign and the achievement of a good result already in a short time. 

Type B Haemophilus Influenzae was perceived by 83.66% of our sample as one of the most severe infections. It is interesting to note that almost half of the participants do not know their vaccination status regarding Type B Haemophilus Influenzae. Type B Haemophilus Influenzae can cause meningitis. The fear of meningitis could explain why, in our sample, all pathogens that can cause this type of manifestation, such as Type B Haemophilus Influenzae, Pneumococcus, and Meningococcus, obtained similar percentages in the perception of severe infection (respectively 83.66%, 83.55%, and 86.18%). In this regard, it is interesting to note the “epidemiological paradox” recorded with other typical childhood pathologies: varicella, measles, mumps, rubella, diphtheria, and pertussis are perceived as pathologies with a low risk of infection, with percentages of 13–20% approximately, and at the same time as the least risky in terms of severity, with percentages of 43–52% approximately. The only exception was represented by diphtheria, perceived as a highly severe disease by 71.4% of the interviewees. Type B Haemophilus Influenzae was instead perceived not only as one of the most severe but also as one of the worst from a risk of infection point of view (33.1%). This could be explained by the general perception of typical childhood pathologies: having survived without significant consequences could make these pathologies perceived as less risky compared to others, including Type B Haemophilus Influenzae, of which subjects knew less about their status. 

Another example is represented by Poliomyelitis: although Europe has been declared Polio-free since 2002 [35], 18.1% of those interviewed declared that they considered themselves at high infectious risk for it. This could be explained by fear of possible cases imported from countries where polio is still endemic or, more simply, by a lack of knowledge about the virus’s circulation in today’s world. 

Vaccine adherence against pneumococcus and seasonal Influenza was shown to be in line with what has been found in other European studies [29,31,36]. Although vaccination adherence against seasonal Influenza was low (33.8%), in this case, it should also be considered that the coincidence with the initial window period of the study could have excluded some adherences. However, even considering those intending to get vaccinated (21.6%), the total is still lower than the PNAIDS target of 70% [28].

No statistically significant differences were found between the STD vaccine group and the remaining group. The percentages of the two groups do not differ in being already vaccinated, willingness to be vaccinated (if not already immune), and perceived low risk of infection (51.2%, 14.8%, and 75.7% for the former and 48.6%, 20.6%, and 76.3% for the latter, respectively), but in perceived high risk of severity (81.7% vs. 65.3%). Although unprotected sex is the most frequent cause of transmission, and in 2021, most new HIV diagnoses in Italy will be attributable to this (83.5%) [26], a significant percentage of subjects (13.6–18.8%) have been reluctant to be vaccinated against these diseases.

Trust in the Italian NHS has encountered conflicting opinions: although almost all the interviewees believe in the information provided by healthcare professionals and in their training and updating on the vaccination subject, there are also other aspects to observe. Furthermore, 40.3% of the sample believes that healthcare workers blame people who do not get vaccinated; 33.3% believe that healthcare professionals give information only on the benefits and not on the vaccine’s risks. One out of four believes that vaccines are an imposition and not a free choice; finally, about 20% believe that there is an economic interest in promoting vaccinations and that vaccination campaigns are not flexible. These views could contribute to increased vaccine hesitancy, and as a result, enhancing patients’ trust in the NHS may improve vaccination adherence. 

This study has some limitations. First, we recruited all the eligible patients followed during the study period who accepted to participate as an opportunistic sample. This may have affected the generalizability of the study. However, data collection and analysis were set up to ensure data quality and exclude participants with ineligibility criteria. Furthermore, no multivariate analysis was performed, which may affect the identification of cause-and-effect links. The main reason is that the sample size is limited and the number of variables is substantial. We have decided to proceed with the multivariate analysis only after continuing the study, which is planned to become multicentric. Analogously, it would be very interesting to identify any potential correlation between each individual vaccination (or vaccination intention) and all the others. However, we deemed it appropriate to wait for the multicenter study to obtain a larger sample size capable of providing more robust results.

## 5. Conclusions

In conclusion, our study was able to highlight a vaccination rate above the European average while at the same time highlighting that there is a significant percentage of subjects who are reluctant towards vaccination. However, the national health system trust analysis of these subjects has highlighted how they believe in healthcare workers and their training but less in the ways in which they interact with patients. It could be that with more informal, transparent, and less guilty communication, it could be possible to counteract vaccine hesitancy in this category of subjects. However, further studies are needed to better understand the attitudes of PLWH and the factors potentially responsible for vaccine hesitancy. 

## Figures and Tables

**Table 1 vaccines-11-01315-t001:** Sample’s vaccination status, % (n) (N = 160).

Pathogenic Agent	Not Vaccinated	Yes, I Am Vaccinated	Natural Immunity	Do Not Know
Hepatitis A	19.61 (30)	48.37 (74)	3.27 (5)	28.76 (44)
Hepatitis B	16.67 (26)	53.21 (83)	4.49 (7)	25.64 (40)
Tetanus (booster)	16.67 (26)	58.33 (91)	0 (0.00)	25.00 (39)
Diphtheria	19.23 (30)	46.79 (73)	0.64 (1)	33.33 (52)
Pertussis	17.42 (27)	50.32 (78)	0 (0.00)	32.26 (50)
HPV	31.17 (48)	44.16 (68)	0 (0.00)	24.68 (38)
Pneumococcus	16.88 (26)	57.14 (88)	0 (0.00)	25.97 (40)
Influenza	48.70 (75)	33.77 (52)	0.65 (1)	16.88 (26)
Varicella	14.94 (23)	30.52 (47)	33.12 (51)	21.43 (33)
Measles	11.61 (18)	62.58 (97)	0 (0.00)	25.81 (40)
Mumps	16.88 (26)	25.97 (40)	24.03 (37)	33.12 (51)
Rubella	18.18 (28)	23.38 (36)	21.43 (33)	37.01 (57)
Poliomyelitis	21.43 (33)	37.66 (58)	1.30 (2)	39.61 (61)
Meningococcus	17.31 (27)	53.21 (83)	0 (0.00)	29.49 (46)
Type B Haemophilus Influenza	29.41 (45)	22.22 (34)	0 (0.00)	48.37 (74)
Herpes Zoster	50.00 (76)	13.16 (20)	7.24 (11)	29.61 (45)

**Table 2 vaccines-11-01315-t002:** Sample’s vaccination intention, % (n) (N = 160).

Pathogenic Agent	No	Yes	Do Not Know	No, I Am Already Immune
Hepatitis A	18.83 (29)	12.34 (19)	19.48 (30)	49.35 (76)
Hepatitis B	13.64 (21)	12.99 (20)	17.53 (27)	55.84 (86)
Tetanus (booster)	13.73 (21)	12.42 (19)	17.65 (27)	56.21 (86)
Diphtheria	18.30 (28)	16.34 (25)	20.26 (31)	45.10 (69)
Pertussis	17.65 (27)	14.38 (22)	20.92 (32)	47.06 (72)
HPV	16.99 (26)	18.95 (29)	24.84 (38)	39.22 (60)
Pneumococcus	11.76 (18)	20.92 (32)	15.03 (23)	52.29 (80)
Influenza	32.03 (49)	21.57 (33)	14.38 (22)	32.03 (49)
Varicella	13.73 (21)	13.07 (20)	15.69 (24)	57.52 (88)
Measles	15.58 (24)	11.69 (18)	18.18 (28)	54.55 (84)
Mumps	18.95 (29)	12.42 (19)	23.53 (36)	45.10 (69)
Rubella	16.99 (26)	15.69 (24)	26.14 (40)	41.18 (63)
Poliomyelitis	20.78 (32)	14.29 (22)	23.38 (36)	41.56 (64)
Meningococcus	14.19 (22)	18.71 (29)	20.00 (31)	47.19 (73)
Type B Haemophilus Influenzae	23.18 (35)	21.85 (33)	33.11 (50)	21.85 (33)
Herpes Zoster	18.18 (28)	39.61 (61)	24.03 (37)	18.18 (28)

**Table 3 vaccines-11-01315-t003:** Risk and severity perceptions of vaccine-preventable infections, % (n) (N = 160).

Pathogenic Agent	Perception of Infection Transmission Risk		Perception of Infection Disease Severity	
	Low Risk	High Risk	Low Severity	High Severity
Hepatitis A	74.84 (119)	25.16 (40)	23.53 (36)	76.47 (117)
Hepatitis B	78.57 (121)	21.43 (33)	12.42 (19)	87.58 (134)
Tetanus (booster)	74.19 (115)	25.81 (40)	16.88 (26)	83.12 (128)
Diphtheria	79.22 (122)	20.78 (32)	28.57 (44)	71.43 (110)
Pertussis	79.87 (123)	20.13 (31)	43.14 (66)	56.86 (87)
HPV	73.55 (114)	26.45 (41)	18.95 (29)	81.05 (124)
Pneumococcus	71.43 (110)	28.57 (44)	16.45 (25)	83.55 (127)
Influenza	50.65 (78)	49.35 (76)	47.06 (72)	52.94 (81)
Varicella	86.93 (133)	13.07 (20)	52.60 (81)	47.40 (73)
Measles	85.62 (131)	14.38 (22)	49.35 (76)	50.65 (78)
Mumps	85.16 (132)	14.84 (23)	45.45 (70)	54.55 (84)
Rubella	82.35 (126)	17.65 (27)	48.34 (73)	51.66 (78)
Poliomyelitis	81.94 (127)	18.06 (28)	21.05 (32)	78.95 (120)
Meningococcus	69.68 (108)	30.32 (47)	13.82 (21)	86.18 (131)
Type B Haemophilus Influenza	66.88 (103)	33.12 (51)	16.34 (25)	83.66 (128)
Herpes Zoster	61.69 (95)	38.31 (59)	29.03 (45)	70.97 (110)

**Table 4 vaccines-11-01315-t004:** Trust in the Italian NHS, % (n) (N = 160).

	I Disagree	I Agree
I believe in the information given by NHS workers	4.38 (7)	95.63 (153)
Healthcare professionals are prepared and up-to-date on vaccines	6.92 (11)	93.08 (148)
Those who do not get vaccinated are blamed by NHS workers	59.75 (95)	40.25 (64)
The organisation of the vaccination offer is flexible in terms of timing and methods	18.87 (30)	81.13 (129)
NHS workers have an economic interest in vaccinations	80.50 (128)	19.50 (31)
NHS workers fail to provide information on vaccine risks	66.67 (106)	33.33 (53)
Vaccines are an imposition, not a free choice	74.84 (119)	25.16 (40)

## Data Availability

The data that support the findings of this study are available from the corresponding author, A.P., upon reasonable request.

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
