# Peer review of "Vaccination and Trust in the National Health System among HIV+ Patients: An Italian Cross-Sectional Survey"

_vaccines, 2023, doi:10.3390/vaccines11081315_

Round 1
Reviewer 1 Report
Thanks for the chance to review this manuscript. It is well written manuscript and I enjoyed reading it.
I have some minor points:
Please add the questionnaire as a supplement. Your previous publication (ref. 18) does not have it as well.
Please add abbreviations for all tables. What do mean by: No, N and n mean?
Table 4: in the last line % in brackets and number outside, please correct.
Table 3: Table title is not really clear. Also, column 2 and 3 titles not clear. Could you please revise?
Why the authors did not do multivariate analysis? I think it will strength the paper. One extra table would do the job.
Ethical issues: Please add the ethical approval number to the Methods. This is a requirement by the STROBE guidelines (studies with human participants). Also, Explain in the Methods who has access to patients identifiers and for how long.
Line 67: the word “a still” confuse the reader. You can remove it.
Line 146: what do you mean by “updating”? Do you mean “updated knowledge”?
Discussion: It was interesting to read about the vulnerable groups in the Introduction. However, it was not discussed in the Discussion. It would be of interest if the author can add a small paragraph about how they compare their findings with different vulnerable groups (see as example 10.3390/vaccines10101634).
Author Response
Thank you very much for the review which allowed us to improve the article and correct several errors!
Please add the questionnaire as a supplement. Your previous publication (ref. 18) does not have it as well.
Unfortunately there appears to be no option to upload the questionnaire as a supplementary file after submission, but we will immediately contact the Academic Editor to have this added. Thank you for your suggestion!
Update: after responding to the reviewers the option to add the supplemental material is back, so we do that immediately!
Please add abbreviations for all tables. What do mean by: No, N and n mean?
Normally with "N" we refer to the size of the total sample while with "n" to that of the sub-sample which in this case answered affirmatively to a certain question. We agree that "No" in table 1 could be misleading, so we changed it to "Not vaccinated"
Table 4: in the last line % in brackets and number outside, please correct.
Corrected!
Table 3: Table title is not really clear. Also, column 2 and 3 titles not clear. Could you please revise?
We agree with that and revised table title and column, hope that is clearer now.
Why the authors did not do multivariate analysis? I think it will strength the paper. One extra table would do the job.
We thank the reviewer for the question about multivariate analysis. Since the sample size is limited, and the number of variables is substantial, we have decided to proceed with the multivariate analysis only after continuing the study, which is planned to become multicentric.
Ethical issues: Please add the ethical approval number to the Methods. This is a requirement by the STROBE guidelines (studies with human participants). Also, Explain in the Methods who has access to patients identifiers and for how long.
We added protocol code and a small sentence in "Data collection" to explain access to the patients identifiers.
Line 67: the word “a still” confuse the reader. You can remove it.
Removed
Line 146: what do you mean by “updating”? Do you mean “updated knowledge”?
Yes! Corrected
Discussion: It was interesting to read about the vulnerable groups in the Introduction. However, it was not discussed in the Discussion. It would be of interest if the author can add a small paragraph about how they compare their findings with different vulnerable groups (see as example 10.3390/vaccines10101634).
We thank the reviewer for this interesting note. Unfortunately we think that the specificity of our investigation does not allow us to compare our results with other groups, except that of HIV positive. If this information is, however, deemed essential, we are certainly willing to further delve into the results.
We really appreciated your contribution, we think it was very helpful in improving our paper. Best regards
Reviewer 2 Report
This is a succinct and generally well-reported cross-sectional study.
There are three changes I would recommend:
1. it's not clear from the rationale really why it was important to do this study. It's suggested that people who are HIV-positive may be (by definition) members of a more socially excluded group, which would tend to be linked to lower vaccination uptake rates. This could be made clearer. It would also be helpful to know whether their HIV status either (a) increases the chance of contracting any of the conditions that are being vaccinated against or (b) increases the likely severity of the condiiton, if contracted. Even if (a) or (b) are not the case, it would be helpful to clarify.
2. Not all potential participants could be recruited to the study, which is understandable, but it would be helpful to give more information on non-recruitment rates (and also on any known factors that increased the chance of non-recruitment). It would also be helpful to have more information (similar to above) on questionnaire items that were not completed.
3. I think the discussion would be easier to read if there was a clearer structure and greater use of sub-headings (if allowed by the journal).
There are a few typos (that would be good to correct) but they don't detract from the overall messages of the article.
Author Response
Thank you very much for the review which allowed us to improve the article!
1) it's not clear from the rationale really why it was important to do this study. It's suggested that people who are HIV-positive may be (by definition) members of a more socially excluded group, which would tend to be linked to lower vaccination uptake rates. This could be made clearer. It would also be helpful to know whether their HIV status either (a) increases the chance of contracting any of the conditions that are being vaccinated against or (b) increases the likely severity of the condiiton, if contracted. Even if (a) or (b) are not the case, it would be helpful to clarify.
We agree with that and added two small sentence in the introduction and two references about point a and b!
2) Not all potential participants could be recruited to the study, which is understandable, but it would be helpful to give more information on non-recruitment rates (and also on any known factors that increased the chance of non-recruitment). It would also be helpful to have more information (similar to above) on questionnaire items that were not completed.
Unfortunately we have no data on the non-recruitment rate, we can only say that compliance was more than good and that it seems the few cases of non-adherence were due to patients' physical problems, language, or difficulties in understanding the questionnaire. There were few non-adherences, but since we didn't have precise data in this regard, we preferred not to specify it. As regards the uncompleted parts of the questionnaire, it seems that most were due to the patient's difficulty in remembering which vaccinations he had received. However, having still recorded good compilation rates in this field with the "I dont know" option, we preferred not to delve into the discussion.
3. I think the discussion would be easier to read if there was a clearer structure and greater use of sub-headings (if allowed by the journal).
We didnt find indications on the guidelines of the journal in this regard. However we agree that the discussion could be difficult to read, so we start a new paragraph to separate the various topics. It should be clearer now.
We really appreciated your contribution, we think it was very helpful in improving our paper. Best regards
Reviewer 3 Report
Review
The authors demonstrated the trust in vaccinations and the national health system among people living with HIV (PLHW) in Italy. The research focus is of important since the PLHW are obvious to be a vulnerable cohort for infectious and pandemic diseases. The introduction is well described to understand the background and aims of their study. The following items should be considered for the revision of manuscript.
1) The correlations of vaccination status between different diseases would appreciate to be shown. For instance, what is the percentage of individuals who have not been vaccinated for Hepatitis A and have also missed other vaccinations? This helps understanding the broader trends of vaccination statuses as shown in Table 1.
2) Similarly, the correlations of vaccination intention between different diseases would be helpful. For example, what is the percentage of individuals who have not intended to be vaccinated for Hepatitis A and have also not intended the other vaccinations?
3) There might be differences in the proportions of infection/severity perception in Table 3 between vaccination intention in Table 2.
4) Minor typo; it would be ’74.84 (119)’ for ‘disagree’ in ‘Vaccines are in imposition’ in Table 4.
5) Correlation between vaccine status and trust in the Italian NHS would help to understand contribution of the trust in NHS for vaccination status.
Author Response
Thank you very much for the review which allowed us to improve the article and correct several errors!
1) The correlations of vaccination status between different diseases would appreciate to be shown. For instance, what is the percentage of individuals who have not been vaccinated for Hepatitis A and have also missed other vaccinations? This helps understanding the broader trends of vaccination statuses as shown in Table 1.
2) Similarly, the correlations of vaccination intention between different diseases would be helpful. For example, what is the percentage of individuals who have not intended to be vaccinated for Hepatitis A and have also not intended the other vaccinations?
Thank you for your advice. It would be very interesting to identify any potential correlation between each individual vaccination (or vaccination intention) and all the others. However, since the vaccinations under consideration are 17, we would end up with approximately 280 combinations, making it challenging to interpret the results due to the limited sample size. Additionally, describing all the findings within the constraints of word and table limits in a scientific article would pose further difficulties. If this information is, however, deemed essential for understanding the phenomenon related to specific individual vaccinations, we are certainly willing to further delve into the results.
3) There might be differences in the proportions of infection/severity perception in Table 3 between vaccination intention in Table 2.
We have checked and everything seems correct.
4) Minor typo; it would be ’74.84 (119)’ for ‘disagree’ in ‘Vaccines are in imposition’ in Table 4.
Except for that, corrected!
5) Correlation between vaccine status and trust in the Italian NHS would help to understand contribution of the trust in NHS for vaccination status.
Thank you for your interesting suggestion. We evaluated the correlation between vaccine status and trust in the Italian NHS and we added the following sentences on the main text:
"Methods: A confidence score towards the NHS (National Health Service) was calculated by assigning one point to responses indicating trust. The total score is the sum of individual scores and can range from 0 to 7, where 7 represents the highest level of trust. To evaluate differences in the confidence score across categories defined by vaccination status, the Kruskal-Wallis or Mann-Whitney test was performed where appropriate.
Results: The confidence score had a median of 6 (IQR=5-7). For most vaccines, there was no significant relationship between vaccination status and the score, except for the following: measles (medians for unvaccinated vs vaccinated were 4.5 vs 6; p=0.027), meningococcus (medians for unvaccinated vs vaccinated were 5 vs 6; p=0.002)."
We really appreciated your contribution, we think it was very helpful in improving our paper. Best regards
Round 2
Reviewer 3 Report
The revised manuscript is now successfully addressed to the reviewers' comments. Further study is warranted to explore the relationship between vaccinations and trust in the national health system globally.